# A plant chitinase controls cortical infection thread progression and nitrogen-fixing symbiosis

Anna Malolepszy[1†‡], Simon Kelly[1†], Kasper Kildegaard Sørensen[2],
Euan Kevin James[3], Christina Kalisch[1§], Zoltan Bozsoki[1], Michael Panting[1],
Stig U Andersen[1], Shusei Sato[4#], Ke Tao[1], Dorthe Bødker Jensen[1], Maria Vinther[1],
Noor de Jong[1], Lene Heegaard Madsen[1], Yosuke Umehara[5], Kira Gysel[1],
Mette U Berentsen[1¶], Mickael Blaise[1**], Knud Jørgen Jensen[2],
Mikkel B Thygesen[2], Niels Sandal[1], Kasper Røjkjær Andersen[1], Simona Radutoiu[1*]

[1]Department of Molecular Biology and Genetics, Aarhus University, Aarhus, Denmark; [2]Department of Chemistry, University of Copenhagen, Frederiksberg, Denmark; [3]The James Hutton Institute, Scotland, United Kingdom; [4]Kazusa DNA Research Institute, Kisarazu, Japan; [5]Institute of Agrobiological Sciences, National Agriculture and Food Research Organization, Tsukuba, Japan

**\*For correspondence:**
radutoiu@mbg.au.dk

[†]These authors contributed equally to this work

**Present address:** [‡]Salk Institute for Biological Studies, La Jolla, United States; [§]Novozymes, Kalundborg, Denmark; [#]Graduate School of Life Sciences, Tohoku University, Miyagi, Japan; [¶]Eurofins Steins Laboratorium A-S, Vejen, Denmark; [**]Institut de Recherche en Infectiologie de Montpellier-IRIM, University of Montpellier, Montpellier, France

**Abstract** Morphogens provide positional information and their concentration is key to the organized development of multicellular organisms. Nitrogen-fixing root nodules are unique organs induced by Nod factor-producing bacteria. Localized production of Nod factors establishes a developmental field within the root where plant cells are reprogrammed to form infection threads and primordia. We found that regulation of Nod factor levels by *Lotus japonicus* is required for the formation of nitrogen-fixing organs, determining the fate of this induced developmental program. Our analysis of plant and bacterial mutants shows that a host chitinase modulates Nod factor levels possibly in a structure-dependent manner. In *Lotus*, this is required for maintaining Nod factor signalling in parallel with the elongation of infection threads within the nodule cortex, while root hair infection and primordia formation are not influenced. Our study shows that infected nodules require balanced levels of Nod factors for completing their transition to functional, nitrogen-fixing organs.

DOI: https://doi.org/10.7554/eLife.38874.001

## Introduction

Molecules of microbial origin have the capacity to induce a morphogenetic response in symbiotic eukaryotic hosts (*Matsuo et al., 2005*; *Alegado et al., 2012*; *Lerouge et al., 1990*). Nitrogen-fixing rhizobia produce species-specific decorated chitin-like molecules, Nod factors (*Lerouge et al., 1990*) that are recognized by LysM receptors in legume roots and initiate two different processes; nodule development and infection thread (IT) formation (*Radutoiu et al., 2003*; *Murakami et al., 2018*). In addition to their signalling capacity, Nod factors have been considered morphogens based on their effect on host developmental programs (*Long, 1996*), but direct measurements of *in planta*-produced Nod factor levels proved intractable, and therefore limited our knowledge on their concentration-dependent effect on the host (*Morieri et al., 2013*). ITs are tubular structures used by bacteria as conduits to advance from the root hair tip towards the root cortex. IT passage through cortical layers involves reprogramming of individual cells for microbial infection (*van Brussel et al., 1992*). Inside nodule primordia, branched IT arrays spread from the main shaft, guiding the confined rhizobia towards competent cortical cells where endocytosis takes place (*Monahan-Giovanelli et al.,*

*2006*). Although a number of components are known (*Xie et al., 2012*; *Yokota et al., 2009*; *Yano et al., 2009*), the mechanisms governing IT formation are not defined, and even less understood is the control of IT branching inside root nodules. Rhizobia grow and multiply during infection thread progression (*Fournier et al., 2008*) and consequently bacterial signals accumulate inside the limited space within ITs. Nevertheless, the host determines the compatibility of Nod factors and exopolysaccharide produced by bacteria during root hair infection, via NFR1/NFR5 and EPR3 receptors, respectively (*Kawaharada et al., 2015*; *Hayashi et al., 2014*).

In parallel with the signalling events induced by Nod factor recognition, legumes express chitinases with Nod factor cleaving activity (*Staehelin et al., 1994*; *Goormachtig et al., 1998*). Plant chitinases are primarily found as components of immune responses induced by chitin producing pathogens (*Lipka et al., 2005*), but novel functions of these enzymes have evolved in plants (*De Jong et al., 1992*) and in symbiotic interactions between eukaryotes and microbes (*Kremer et al., 2013*). The identification of Nod factor-cleaving chitinases induced during legume-rhizobia symbiosis led to a model where these enzymes are required for Nod factor hydrolysis to avoid activation of immune responses induced by these chitin-like molecules during symbiosis (*Goormachtig et al., 1998*; *Savoure et al., 1997*; *Staehelin et al., 2000*). However, unlike chitin, Nod factors do not trigger immune responses in *Lotus* (*Bozsoki et al., 2017*) and in this context the function of Nod factor cleaving chitinases remained unclear. This was further confirmed by the *Medicago truncatula*, Nod factor hydrolase (NFH1) mutants that display delayed infection thread formation and nodule hypertrophy (*Cai et al., 2018*).

## Results

We identified *chit5-1*, *chit5-2* and *chit5-3* as fix minus Gifu mutant alleles from screens aimed at discovering symbiotic defective mutations following inoculation with *Mesorhizobium loti* (*Sandal et al., 2006*) (*Figure 1A*). The impaired symbiotic association was reflected in significantly reduced shoot growth (*Figure 1—figure supplement 1*) and nitrogen deficiency symptoms when compared to wild type (*Figure 1A*). This defective phenotype was alleviated by the addition of nitrogen (10 mM KNO$_3$) to the growth substrate (*Figure 1—figure supplement 2*). Detailed analysis of the symbiotic phenotype manifested by the three alleles revealed that nodule organogenesis was initiated in the presence of *M. loti* (*Figure 1—source data 1*), but most nodules remained arrested at the primordia stage with only a few developing into large, white or pink-spotted nodules (*Figure 1B,C*). Assessment of bacterial nitrogenase activity through acetylene reduction assays revealed that mutant nodules, despite their occasional functional appearance (pink-spotted), had significantly reduced nitrogenase activity compared to wild-type (*Figure 1D*). The limited nitrogen fixation capacity is reflected in the observed plant growth and nodule phenotypes. This was further supported by reduced transcriptional activation of leghemoglobin (*Lb3*) and sulfate transporter (*Sst1*) genes which are prominent molecular markers for mature nodules and successful nitrogen-fixing symbiosis (*Ott et al., 2005*; *Krusell et al., 2005*) (*Figure 1—figure supplement 3*). In contrast, transcripts associated with early rhizobial infection, early nodulin *N6* (*Mathis et al., 1999*), pectate lyase *Npl1* (*Xie et al., 2012*) and transcription factor *Ern1* (*Kawaharada et al., 2017a*), were induced to higher levels in the *chit5* mutants compared to wild-type plants (*Figure 1—figure supplement 3*). No morphological signatures associated with activation of plant defense responses, such as the onset of early senescence, previously reported for some fix minus mutants (*Bourcy et al., 2013*) was observed in *chit5* plants (*Figure 1B*). *Mpk3*, *Wrky29* and *Wrky*33, marker genes upregulated during plant immunity (*Lohmann et al., 2010*), were found to have a wild-type level of expression in both uninoculated and 21-dpi whole mutant roots (*Figure 1—figure supplement 3* and *Figure 1—source data 2*). Recent transcriptome analyses of *Lotus* (*Kelly et al., 2018a*) revealed that progression of symbiosis in wild-type roots and formation of the symbiotic nodules (21 dpi) is accompanied by downregulation of these immunity marker genes (*Figure 1—figure supplement 4A*). We have analysed the expression of these marker genes in 21dpi nodules of wild-type and *chit5* mutants. We found a higher level of the *Npl1*, *Mpk3*, *Wrky29* and *Wrky33* transcripts in *chit5* mutant nodules compared to wild-type nodules (*Figure 1—figure supplement 4*). The differences in the expression levels of the early symbiotic and defense markers detected between wild-type and *chit5* mutants indicate a deregulated symbiotic and immune signalling, seemingly arresting nodules at an early infection stage. We investigated whether rhizobial infection was affected and found that a normal

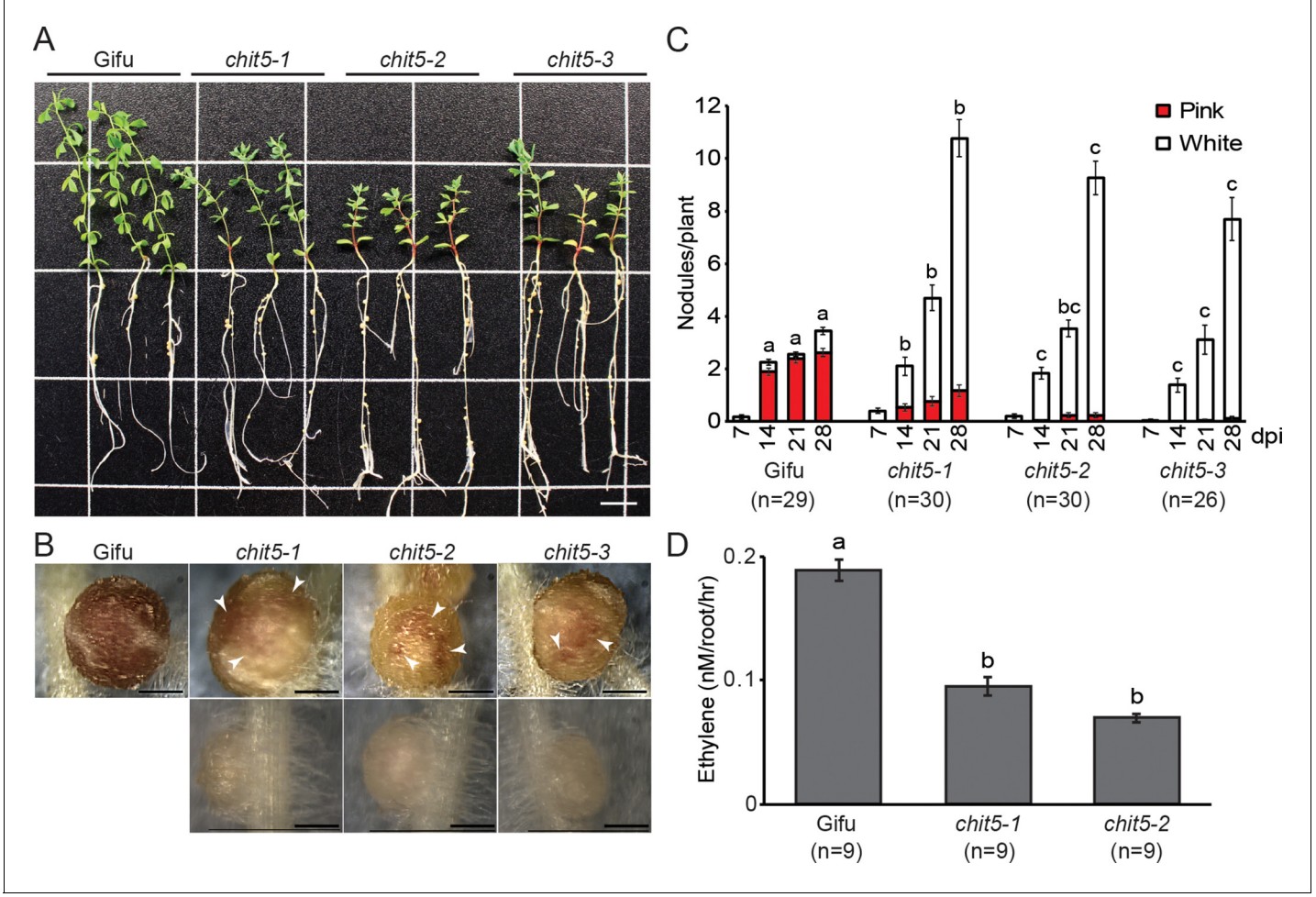

**Figure 1.** *chit5* mutants are defective in nitrogen-fixing symbiosis. (**A**) Three representative plants of wild-type Gifu and *chit5* mutant alleles 6 weeks post-inoculation with *M. loti* R7A. Scale bar is 1 cm. (**B**) Gifu plants form pink nodules whilst *chit5* mutants form mainly small white nodules, and occasionally pink-spotted (white arrows) nodules. Scale bars are 0.5 cm. (**C**) Average number of pink and white nodules formed by *M. loti* R7A over 4 weeks. (**D**) Nitrogenase activity measured as the amount of ethylene produced from acetylene in nodules of Gifu and *chit5* mutants inoculated with *M. loti* R7A. (**C**) and (**D**) Error bars represent SEM and statistical comparisons of the number of pink nodules formed between genotypes at each time point are shown using ANOVA and Tukey post hoc testing with p values < 0.01 as indicated by different letters.

DOI: https://doi.org/10.7554/eLife.38874.002

The following source data and figure supplements are available for figure 1:

**Source data 1.** Bacterial strains used in this study.
DOI: https://doi.org/10.7554/eLife.38874.007
**Source data 2.** Primers used for RT-qPCR analyses.
DOI: https://doi.org/10.7554/eLife.38874.008
**Figure supplement 1.** Shoot length of Gifu and *chit5* mutants following inoculation with *M.loti* R7A.
DOI: https://doi.org/10.7554/eLife.38874.003
**Figure supplement 2.** Wild-type and *chit5-3* mutant show similar phenotype when grown in nitrogen-supplemented soil.
DOI: https://doi.org/10.7554/eLife.38874.004
**Figure supplement 3.** RT-qPCR analysis of symbiotic and defense gene expression in whole roots.
DOI: https://doi.org/10.7554/eLife.38874.005
**Figure supplement 4.** Expression of symbiotic and defense gene during symbiosis (A) and in wild-type and *chit5* nodules (B).
DOI: https://doi.org/10.7554/eLife.38874.006

number of ITs were formed in *chit5* mutant root hairs (**Figure 2A**), and these proceeded in a manner similar to wild-type plants (**Figure 2—figure supplement 1**). Together, these results demonstrate that the early programs initiated by Nod factor signalling, nodule organogenesis and formation of root hair ITs, operate normally in *chit5* mutants.

In determinate nodules, like those formed by *L. japonicus*, ITs are present inside the cortex of mature nodules and can be quantified (**Madsen et al., 2010**). We have performed detailed investigations of IT phenotypes in wild-type and *chit5* mutant nodule sections (n = 30) using light and electron microscopy. Counting of the cortical ITs formed inside wild-type and infected mutant nodules revealed a severe and significant reduction in their number in the mutant nodules (**Figure 2B**). Mutants displayed an impaired symbiotic occupancy of the nodule central tissue, with only scattered infected cells that formed inside the pink-spotted nodules (**Figure 2C**). Bacteria accumulated in the intercellular spaces between nodule cells (**Figure 2C and D**), giving rise to fewer, localized intracellular infections (**Figure 2D**). Symbiosomes present in the infected mutant cells contained multiple bacteroids of irregular shape and of atypical appearance (accumulation of white matrix) (**Figure 2D**). These phenotypes are likely a result of symbiotic and defense genes deregulation inside *chit5* nodules. These results demonstrate that CHIT5 functions during cortical IT development and branching within nodules, and is required for the establishment of fully functional nitrogen-fixing symbiosis.

Map-based cloning and whole genome sequencing revealed that all three alleles are defective in one of the three chitinase-coding genes present at the identified genomic location (**Figure 3A**, **Figure 3—figure supplement 1**, and **Figure 3—figure supplement 2**), while the remaining two paralogs are pseudogenes with premature stop codons (**Figure 3—figure supplement 2**).The *chit5-1* and *chit5-3* mutants were found to have large genomic deletions encompassing the chitinase gene, while *chit5-2* carries a point mutation (CCA to CTA), that results in a proline to leucine ($P_{168}$-L) transition in the predicted protein sequence (**Figure 3A** and **Figure 3—figure supplement 2**). The symbiotic defective phenotype of all three alleles was restored by genetic complementation with the wild-type *Chit5* gene expressed from its native promoter (2072 bp) and terminator (2108 bp) regions, or from the *LjUbiquitin* promoter and *Nos* terminator (**Figure 3—figure supplement 3**). Analyses of *Chit5* expression using transcriptional reporters (*Chit5*-tYFPnls and *Chit5*-GUS) revealed promoter activity in all cells of uninoculated roots, in infected root hairs (**Figure 3B**), and nodule primordia (**Figure 3—figure supplement 4**). *Chit5* promoter activity decreased inside fully functional nodules, indicating a reduced requirement for *Chit5* at later stages of a functional symbiosis (**Figure 3B**, **Figure 3—figure supplement 4**).

The CHIT5 protein consists of a predicted signal peptide at the amino terminus followed by a class V (glycosyl hydrolase 18) domain with a conserved catalytic DxDxE motif (**Figure 3—figure supplement 5**). The proline residue mutated in *chit5*-2 allele is located immediately after the catalytic site, and is conserved across GH18-type chitinases, indicating a crucial role for the function of this protein (**Figure 3—figure supplement 5**). CHIT5 shares a high level of similarity to *Medicago truncatula* NFH1 (**Zhang et al., 2016**), including the presence of A and B loops and the proline residue ($P_{294}$), previously reported as essential for Nod factor hydrolysis (**Zhang et al., 2016**). Furthermore, we found that a version of CHIT5 mutated in the DxDxE motif ($E_{166}$ to K) was no longer able to complement the symbiotic defective phenotype, suggesting that enzymatic activity is required for CHIT5 function *in planta* (**Figure 3—figure supplement 6**). To investigate the enzymatic properties of CHIT5 *in planta*, we took advantage of its predicted secretion (**Figure 3—figure supplement 5**) and ubiquitous expression in *Lotus* roots (**Figure 3B**) and analysed the hydrolytic capacities of wild-type and mutant root exudates on chitin hexamers (hexa-N-acetylchitohexaose, abbreviated CO-VI) and on R7A Nod factor (NodMI-V(C18:1, Me, Cb, FucAc), abbreviated LCO-V).

For testing CHIT5-dependent hydrolysis of CO-VI by root exudates we made use of a physiological test based on the differential capacity of CO-VI, CO-III/CO-II and Nod factor elicitors to induce ROS production in *Lotus* (**Bozsoki et al., 2017**). CO-VI elicitors present in the control samples (not exposed to plant root exudates) induced ROS production, while those exposed to wild-type or mutant roots lost this capacity (**Figure 3C**). This shows that CO-VI hydrolysis took place during incubation with plant roots irrespective of the genotype and that CHIT5 is thus not solely required for CO-VI hydrolysis in the *Lotus* rhizosphere. Similar analyses performed with LCO-V showed no ROS production, confirming that intact R7A Nod factor molecules (present in the control samples) or their possible hydrolysis products resulting from exposure to plant roots, lack the ability to induce ROS in *Lotus* roots (**Figure 3D**) (**Bozsoki et al., 2017**). Analyses of butanol-extracted fractions from LCO-V

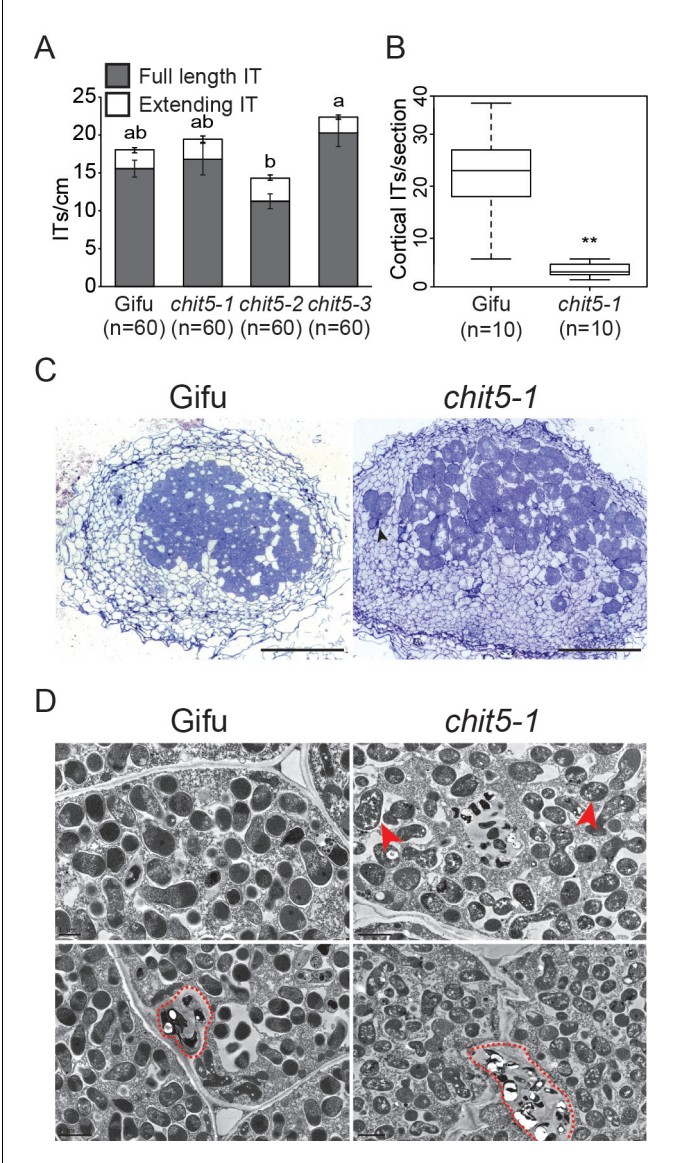

**Figure 2.** *chit5* mutants have a defective nodule infection phenotype. (**A**) Infection threads induced by *M. loti* + DsRed in Gifu and *chit5* mutant root hair at 10 dpi . Error bars represent SEM and statistical comparisons of the number of full length infection threads formed between genotypes at each time point are shown using ANOVA and Tukey post hoc testing with p values < 0.01 as indicated by different letters. (**B**) Box plot of cortical infection thread counts performed on Gifu and *chit5-1* nodule sections. *t* test *p* values are indicated by asterisks (**<0.01). (**C**) Light microscopy of nodule sections from the indicated plant genotypes 4 wpi stained with toluidine blue. Black arrows indicate bacteria between nodule cells. Scale bars are 200 µm (**D**) Transmission electron microscopy of nodule sections from the indicated plant genotypes 4 wpi. Red arrows point out bacteroids of stressed appearance with accumulated white spots, red dashed lines outline infection threads or invagination from the intercellular space. Note the difference between PHB bodies (large, white spots) characteristic for *Mesorhizobium* when present inside infection threads (red dashed outlines), and the small white spots present in the symbiosomes formed in the *chit5* mutant nodules (red arrows). Scale bars are 2 µm. Corresponding images of *chit5-2* and *chit5-3* are in *Figure 2—figure supplement 1*.

DOI: https://doi.org/10.7554/eLife.38874.009

The following figure supplement is available for figure 2:

**Figure supplement 1.** Infection phenotype of wild-type and *chit5* mutants.
DOI: https://doi.org/10.7554/eLife.38874.010

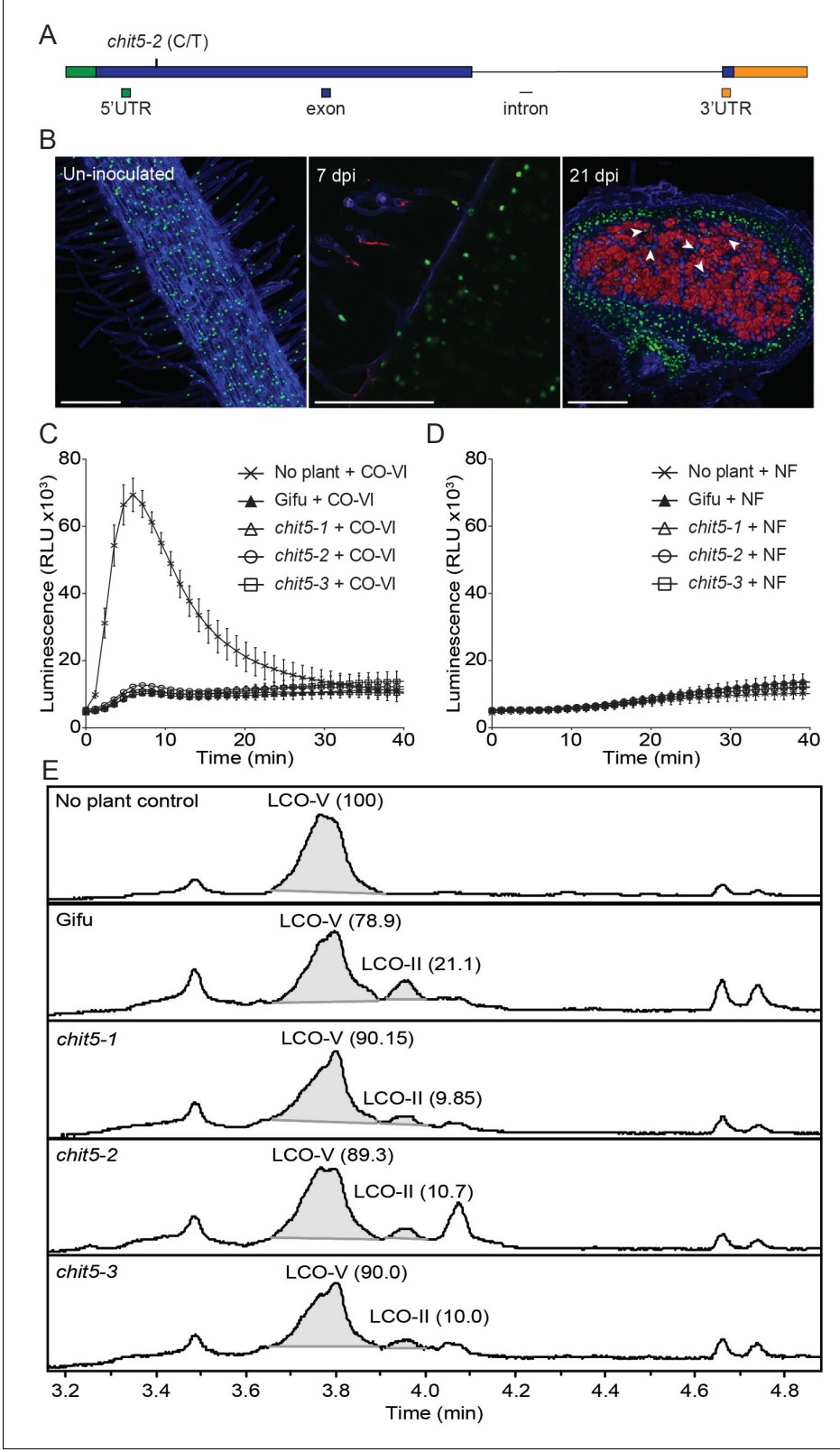

**Figure 3.** *Chit5* encodes a root expressed class V chitinase with Nod factor hydrolase activity. (**A**) *Chit5* gene structure. Mutation in *chit5-2* allele is shown. *Chit5* gene is deleted in *chit5-1* and *chit5-3* alleles. (**B**) *Chit5* promoter activity was monitored in Gifu roots transformed with a *Chit5* promoter-tYFP-NLS (green nuclei) in uninoculated and *M. loti* R7A + DsRed inoculated roots. White arrows highlight examples of uninfected cells

*Figure 3 continued on next page*

*Figure 3 continued*

showing *Chit5* promoter activity. Scale bars are 200 µm. (**C**) ROS induced by exudates of CO-VI-treated plants. (**D**) Absence of ROS induction by exudates of *M. loti* R7A Nod factor-treated plant exudates. (**E**) *M. loti* R7A Nod factor hydrolysis in the presence of the indicated plant genotypes measured by HPLC-MS. The average relative percentage of LCO-V and LCO-II fractions from two biological replicates (***Figure 3—source data 1***) determined from peak-peak integration is indicated in brackets. The peak eluting near 4.1 min is present in all plant treated samples but was not detected in the control sample. It lacks the characteristic fragmentation pattern of LCOs, and appears to be an aromatic, small molecule unrelated to chitinase activity.
DOI: https://doi.org/10.7554/eLife.38874.011

The following source data and figure supplements are available for figure 3:

**Source data 1.** HPLC-MS analysis of Nod factor isolated after exposure to roots of wild-type Gifu or *chit5* mutants.
DOI: https://doi.org/10.7554/eLife.38874.017
**Figure supplement 3.** Complementation of *chit5* mutant alleles with *Chit5.*
DOI: https://doi.org/10.7554/eLife.38874.012
**Figure supplement 4.** *Chit5* promoter-GUS reporter analysis .
DOI: https://doi.org/10.7554/eLife.38874.013
**Figure supplement 5.** Amino acid alignment of CHIT5, AtCHIC, MtNFH1.
DOI: https://doi.org/10.7554/eLife.38874.014
**Figure supplement 6.** Complementation of *chit5* mutant alleles with *Chit5* (E166–K).
DOI: https://doi.org/10.7554/eLife.38874.015
**Figure supplement 7.** Several *Lotus japonicus* chitinase genes are expressed in roots.
DOI: https://doi.org/10.7554/eLife.38874.016
**Figure supplement 2.** Alignment of *Chit5* paralogs.
DOI: https://doi.org/10.7554/eLife.38874.026
**Figure supplement 1.** Map-based cloning of *Chit5* gene.
DOI: https://doi.org/10.7554/eLife.38874.027

samples determined that hydrolysis of the R7A Nod factor into NodMI-II (C18:1, Me, Cb, abbreviated LCO-II) was induced by exudates from roots (***Figure 3E***). Exudates from all *chit5* mutants induced lower production of LCO-II than wild-type exudates, and there was a corresponding reduction in LCO-V degradation by mutants compared with wild-type (***Figure 3E*** and ***Figure 3—source data 1***). Since LCO-II was produced after exposure to *chit5* exudates, additional plant chitinases expressed in *Lotus* roots (***Figure 3—figure supplement 7***) may contribute to Nod factor hydrolysis in the *Lotus* rhizosphere. Our *in planta* assays demonstrate that secreted CHIT5 contributes to hydrolysis of R7A Nod factor. Nevertheless, CHIT5 and unknown hydrolase(s) induce only a limited hydrolysis of the fully compatible Nod factor, most likely to ensure that symbiotic signalling occurs, as shown in wild-type plants.

*M. loti* NodD1 and NodD2 transcriptional regulators control the expression of Nod factor biosynthesis genes and are preferentially active at distinct stages of *M. loti-Lotus* symbiosis (***Kelly et al., 2018b***). NodD1 is primarily active inside root hair ITs, while NodD2 activates the transcription of Nod factor biosynthesis genes within nodules. NodD1 and NodD2 were found to differ significantly in their capacity to induce Nod factor biosynthesis genes transcription on *Lotus* roots and coordinated activity between them ensures an intermediate gene expression in wild-type R7A (***Kelly et al., 2018b***). Our analysis of *chit5* mutants revealed a requirement of CHIT5 for cortical IT extension leading to effective colonization of nitrogen-fixing nodules (***Figure 2***). This provided us with the opportunity to further investigate the necessity of CHIT5 for monitoring Nod factor levels *in planta*, by assessing the phenotypes of *nodD1* ($D1^-/D2^+$) and *nodD2* ($D1^+/D2^-$) bacterial mutants. *chit5* mutants inoculated with *nodD1*, that induces a 3-fold higher level of Nod factor biosynthesis gene transcription compared to wild-type R7A in response to *Lotus* root exudates (***Kelly et al., 2018b***), displayed a severe nitrogen-deficient phenotype (***Figure 4A***). Comparable root hair IT formation was observed on wild-type and *chit5* plants (***Figure 4—figure supplement 1***) yet nodules induced by *nodD1* on *chit5* mutants were even more severely impaired than those induced by R7A. Nodules were rarely or superficially infected and were found to be severely ineffective in acetylene reduction assays (***Figure 4*** and ***Figure 4—figure supplement 1***). In contrast, inoculation with *nodD2*, that induces a 4-fold lower level of Nod factor biosynthesis gene transcription compared to wild-type R7A

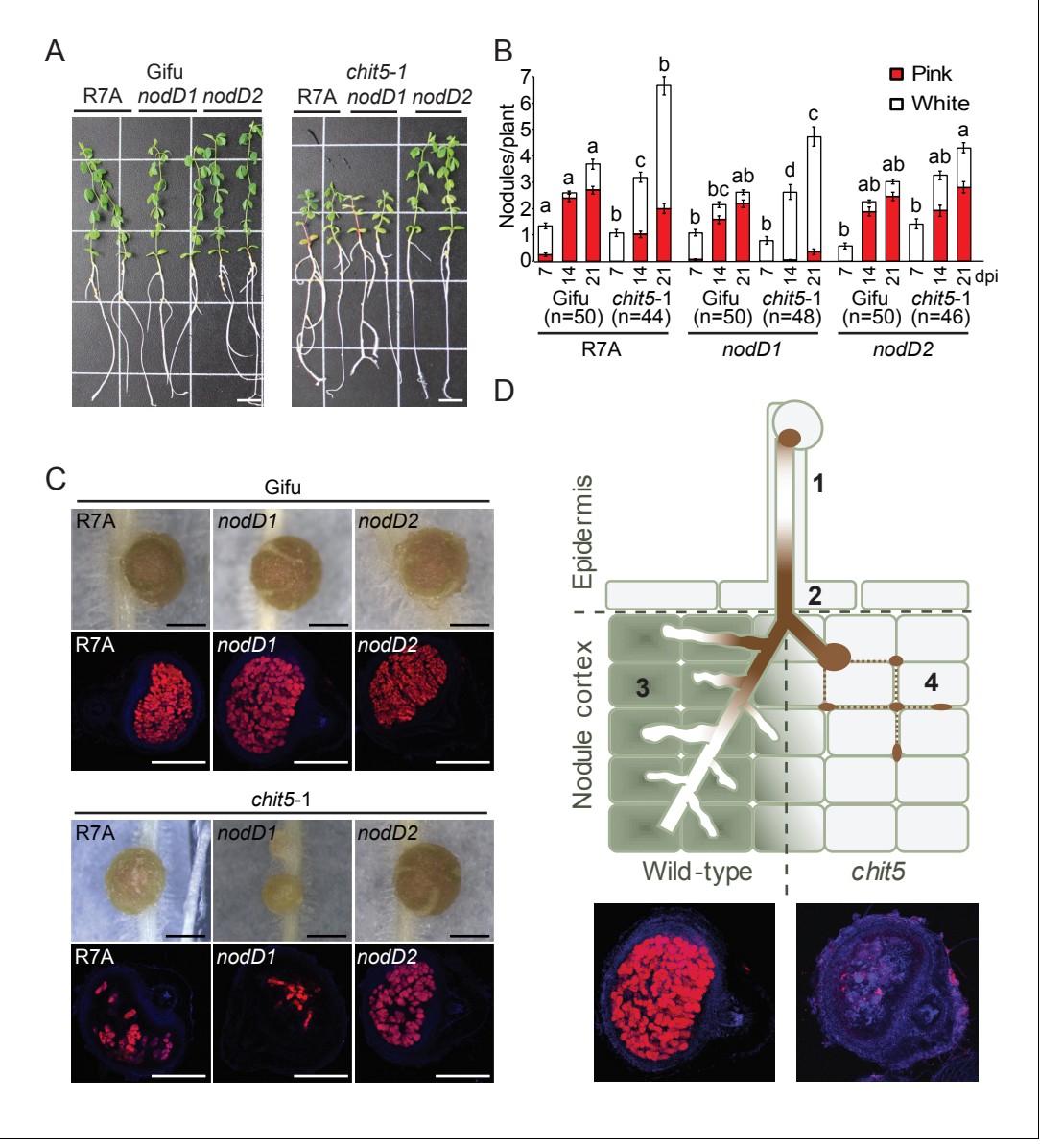

**Figure 4.** The symbiotic impairment of *chit5* mutants can be overcome by an *M.loti* R7A mutant affected in the regulation of Nod factor biosynthesis. (**A**) Representative images of Gifu and *chit5-1* plants inoculated with the indicated *M. loti* R7A strains 7 wpi. Scale bars are 1 cm. (**B**) Average number of pink and white nodules formed on Gifu and *chit5-1* by the indicated *M. loti* R7A strains over 3 weeks. Error bars represent SEM and statistical comparisons of the number of pink nodules formed between strains on each genotype at each time point are shown using ANOVA and Tukey post hoc testing with p values < 0.05 as indicated by different letters. (**C**) Light microscopy of whole nodules 3 wpi and confocal images of nodule sections from Gifu and *chit5-1* plants inoculated with the indicated strains tagged with DsRed. Scale bars are 0.5 cm. (**D**) Proposed model of CHIT5 activity and representative nodule images of wild-type and *chit5* mutant illustrating the observed phenotype. (1) Bacteria maintain a low level of Nod factors within root hair-traversing ITs by preferential activity of NodD1. (2) and (3) Bacterial amplification inside ITs, coupled with the switch for preferential activity of NodD2 leads to higher levels of fully decorated pentameric Nod factors (LCO-V). *Chit5* expression (green filled cells in wild-type) is crucial for maintaining a balanced Nod factor level enabling IT extension inside primordia, efficient bacterial endocytosis and, ultimately, development of the nitrogen-fixing organ (3). (4) In the *chit5* mutants (light grey filled cells) higher levels of Nod factors (LCO-V) impede IT elongation and branching inside primordia leading to bacteria accumulation in between the cells (dotted brown line) and scattered infection.

DOI: https://doi.org/10.7554/eLife.38874.018

The following figure supplements are available for figure 4:

*Figure 4 continued on next page*

*Figure 4 continued*

**Figure supplement 1.** *chit5* mutant phenotypes with *M.loti* R7A mutants affected in the regulation of Nod factor biosynthesis.
DOI: https://doi.org/10.7554/eLife.38874.019

**Figure supplement 2.** Overexpression of *Nfr1* and *Nfr5* does not rescue the symbiotic impairment of *chit5-1*.
DOI: https://doi.org/10.7554/eLife.38874.020

**Figure supplement 3.** *chit5* mutant phenotypes with *M.loti* R7A mutants affected in the biosynthesis of fully decorated Nod factor.
DOI: https://doi.org/10.7554/eLife.38874.021

**Figure supplement 4.** *chit5* mutants show a nitrogen-deficient phenotype when grown in soil.
DOI: https://doi.org/10.7554/eLife.38874.022

**Figure supplement 5.** Genomic location and expression pattern of *MtNfh1* and the two close-located paralogs.
DOI: https://doi.org/10.7554/eLife.38874.023

**Figure supplement 6.** GH18 proteins from *M.truncatula* and *L.japonicus*.
DOI: https://doi.org/10.7554/eLife.38874.024

**Figure supplement 7.** Sequencing traces from PCR products using genomic DNA (gDNA), cDNA from water treated roots (R_M), *M.loti* inoculated roots-3dpi (R_3dpi),−10dpi (R_10dpi) and nodules-21 dpi (N_21dpi) as template.
DOI: https://doi.org/10.7554/eLife.38874.025

(*Kelly et al., 2018b*), resulted in full restoration of *chit5* defective phenotypes (*Figure 4* and *Figure 4—figure supplement 1*). A similar number of pink nodules were formed on wild-type and *chit5* mutant roots and plants were no longer nitrogen-deficient. Fully colonized nodules with efficient nitrogen-fixing symbiosomes were formed (*Figure 4—figure supplement 1*). These contrasting phenotypes of *chit5* mutants in response to *M. loti* strains affected in the regulation of Nod factor biosynthesis indicate that CHIT5-dependent modulation of Nod factor levels within nodules is critical for cortical IT development and complete transition to nitrogen-fixing symbiosis.

Nod factors are perceived by the NFR1/NFR5 receptors (*Radutoiu et al., 2003*). The expression of *Chit5* is similar to that of *Nfr1/Nfr5* (*Kawaharada et al., 2017b*) and follows the spatio-temporal development of symbiosis in the root and nodules (*Figure 3B* and *Figure 3—figure supplement 4*). The observed requirement for tight modulation of Nod factor levels inside nodules (*Figure 4A* and *Figure 4—figure supplement 1*) prompted us to investigate whether the *chit5* phenotype in the presence of wild-type *M. loti* could reflect a bottleneck in Nod factor signalling due to limited/tightly controlled NFR-dependent signalling. We tested this possibility by overexpressing *Nfr1* and *Nfr5* receptors in *chit5* mutants, but observed no change in their symbiotic phenotype after inoculation with *M. loti* R7A (*Figure 4—figure supplement 2*). This indicates that increased and ectopic expression of Nod factor receptors is insufficient to restore the *chit5* defective phenotype.

Bacterial genetics coupled with *in planta* phenotypic studies and chemical analyses of Nod factors produced by rhizobial mutants have determined that various Nod factor decorations are required for initiation and progression of symbiosis (*Ehrhardt et al., 1995*; *Rodpothong et al., 2009*). We investigated the importance of Nod factor decorations for CHIT5 activity in *Lotus* by assessing the phenotypes induced by R7A *nolL* and *nodZ* mutants that produce Nod factors lacking acetyl (LC (C18:1, Me, Cb, Fuc)) and acetylated-fucosyl decorations (LCO-V(18:1, Me, Cb)), respectively on the reducing-end of the Nod factor (*Rodpothong et al., 2009*). Compared to wild-type *M. loti*, these bacterial mutants are delayed in initiating nodule organogenesis and form fewer infected nodules (*Figure 4—figure supplement 3*), a phenotype similar to the one induced by *M. loti nodD1* mutant (*Kelly et al., 2018b*). Interestingly, we found that these bacterial mutants infected *chit5* mutants and wild-type similarly and that, compared to wild-type or *M. loti nodD1*, no significant differences in nodule number, infection rate and nitrogen fixing ability were observed between plant genotypes (*Figure 4—figure supplement 3*). This suggests that cortical infection and onset of nitrogen fixation in the symbiotic organ might require tight modulation of Nod factor levels in a structure-dependent manner. Decorations have been shown to be important for determining the rate of Nod factor degradation by different chitinases (*Ovtsyna et al., 2000*). Nod factors produced by *nodZ* and *nolL* mutants may become accessible to other chitinases present in the *chit5* mutants for degradation. Alternatively, Nod factor signalling induced by *nodZ* and *nolL* mutants may be reduced compared to

wild-type bacteria and therefore not requiring CHIT5-dependent modulation. These findings based on analyses of plant and bacterial mutants in binary interactions questioned the functional relevance of CHIT5 in the context of natural environments where a diverse rhizobial population, presumably producing Nod factors with various structures at various levels, is present. We investigated this by analysing the phenotypes of *chit5* mutant plants grown in soil and exposed to the natural rhizobial population compared with wild-type and *nfr5-2* plants. We found that both *chit5* and *nfr5* mutant plants were clearly distinctive from wild-type, as they were nitrogen starved in spite of numerous nodule primordia formed on *chit5* plants (*Figure 4—figure supplement 4*). This indicates, that at least in the presence of the tested soil rhizobial population, CHIT5 is a major determinant for the onset of nitrogen-fixing symbiosis in the nodules whose formation is controlled by NFR5.

## Discussion

Our study revealed that in *L. japonicus*, CHIT5-dependent hydrolysis of the Nod factor morphogen plays a crucial role during primordia infection and for full-transition to symbiotic nitrogen fixation. *Chit5* in *L. japonicus* and *Nfh1* in *M. truncatula* are highly similar, however the mutant phenotypes in the two model-legumes are very different (*Cai et al., 2018*). Unlike in the *nfh1* mutant (*Cai et al., 2018*), the number and appearance of root hair ITs and the size of nodule primordia appear not to be affected in *chit5* mutants, indicating that in *Lotus* other mechanisms independent of CHIT5 control these early stages of symbiosis. *Chit5* and *Nfh1* are part of genomic clusters with closely-located paralogs (*Figure 3—figure supplement 1*, *Figure 4—figure supplements 5* and *6*). In *Lotus*, two of the three paralogs evolved as pseudogenes (*Figure 3—figure supplement 2* and *Figure 4—figure supplement 7*), while in *Medicago* all are functional genes and expressed (*Figure 4—figure supplement 5*). This may contribute to the differential phenotypes observed in the two model legumes, and may explain the strong phenotype displayed by *Chit5* mutants in *Lotus*. Alternatively, *Chit5* and *Nfh1* might be highly similar but non-orthologous genes. Based on our observations from *Lotus* we propose a likely scenario explaining CHIT5 function (*Figure 4D*). Bacteria maintain a low level of Nod factors within root hair-traversing ITs by preferential activity of NodD1 (*Kelly et al., 2018b*). Bacterial amplification inside primordia-elongating ITs, coupled with the switch for preferential activity of NodD2 (*Kelly et al., 2018b*) leads to higher levels of fully decorated pentameric Nod factors (LCO-V) that unbalance the symbiotic and immune signalling in the cortical cells. CHIT5 hydrolytic activity on LCO-V is crucial for maintaining balanced symbiotic and defense signalling in the root developmental field and is required for cortical IT extension inside primordia, efficient bacterial endocytosis and ultimately, development of the nitrogen-fixing organ. In the absence of *Chit5*, IT elongation is impaired leading to bacteria accumulation in the intercellular space and sparse intracellular infection (*Figure 4D*). However, we cannot exclude that a CHIT5-dependent product derived from Nod-factor hydrolysis plays a role during nodule infection. Interestingly, similar but less frequent infections, as observed in *chit5* mutants inoculated with wild-type *M. loti*, was reported for spontaneous nodules formed on *nfr1snf1* plants when infected by an *M. loti nodC* mutant lacking the ability to produce Nod factors. Surprisingly, the symbiosomes formed by the *M. loti nodC* mutant were reported to have normal appearance and the plants to be nitrogen proficient (*Madsen et al., 2010*). These results suggest that Nod factors, or their hydrolytic products, are dispensable for nitrogen fixation within infected cells. Based on these results and our findings from symbiotic gene expression in *chit5* mutants and their phenotype in the presence of mutant rhizobia, we consider the scenario proposed in *Figure 4D* as the most likely framework for CHIT5 function. Future studies will likely reveal which molecular determinants are responsible for decoding the CHIT5 output into nitrogen-fixing state inside infected cells.

## Materials and methods

### Plant materials

*Lotus japonicus* ecotype Gifu B-129 (*Handberg and Stougaard, 1992*) was used as the wild-type plant. Plants were grown at 21°C with 16 h day and 8 hr night cycles. *Agrobacterium* strain AR1193 (*Stougaard et al., 1987*) was used for hairy-root transformation experiments, carried out as

described previously (*Petit et al., 1987*). Plants were inoculated with corresponding bacterial suspensions, $OD_{600}$ = 0.02.

## Bacterial strains and plasmids

Bacterial strains and plasmids used in this study are listed in *Figure 1—source data 1*. *Mesorhizobium loti* R7A (*Sullivan et al., 2002*; *Kelly et al., 2014*) and mutant strains were cultured at 28°C in YMB or G/RDM medium (*Ronson et al., 1987*). Strains expressing fluorescent reporters were constructed by the introduction of pSKDSRED or pSKGFP (*Kelly et al., 2013*). Antibiotics were added to media as required at the following concentrations: tetracycline, 2 µg ml$^{-1}$; rifampicin, 100 µg ml$^{-1}$; spectinomycin, 100 µg ml$^{-1}$; ampicillin, 100 µg ml$^{-1}$.

## Chit5 gene cloning

The *chit5-1* (*sym43*), *chit5-2* (*sym103-1*) and *chit5-3* (*sym103-2*) mutants were isolated as fix⁻ mutants in the background of *L. japonicus* ecotype 'Gifu', and they were described by *Sandal et al. (2006)*. An F2 mapping population was established by crossing *chit5-1* mutant to wild-type *L. japonicus* ecotype 'MG-20'. In total, 1508 F2 offspring mutant plants allowed us to limit the region of interest to 483 kb on chromosome 5 (*Figure 3—figure supplement 1*). Additional attempts to further limit the candidate region were not successful due to suppression of recombination in this chromosomal region. Next, we used a combination of BAC/TAC subcloning and whole genome sequencing of nuclear DNA (*Schneeberger et al., 2009*) from the three alleles by Illumina technology (NexTera Library Kit) to pinpoint the candidate mutated gene. SHOREmap strategy (*Schneeberger et al., 2009*) was used for SNP calling based on *L. japonicus* v.3.0 genome. We identified that *chit5-1* and *chit5-3* contain deletions of app. 13 and 9 kb, respectively in this region while *chit5-2* has a point mutation C/T in a predicted chitinase gene. Three chitinases (*Chit5, Chit5a* and *Chit5b*) sharing high identity in the coding, 5' and 3' DNA flanking regions were identified to be present in this genomic region (*Figure 3—figure supplement 2*). *Chit5a* and *Chit5b* have early stop codons and are therefore pseudogenes (*Figure 3—figure supplement 2*). The *chit5-1* and *chit5-3* mutants lack *Chit5* gene, and one of the pseudogenes, while *chit5-2* has all three genes, but contains a point mutation (C/T) in *Chit5*. The precise location of the three genes in the current version of *L. japonicus* genome is not possible to assign due to the high level of similarity and repetitive nature of the DNA sequences present in this region.

## Plant phenotypic assessment

Seed sterilization and plant-growth setups for nodulation and infection thread assays were as previously described (*Kawaharada et al., 2015*). For nodulation assays plants were inoculated with R7A, *nodD1*, *nodD2*, *nolL* and *nodZ* and scored weekly. For IT counts, plants were inoculated with strains carrying the pSKDSRED reporter plasmid. Roots of 10 dpi- plants (n = 20 per biological replica) were cut into 1 cm pieces and 20 pieces were examined for infection thread counts. For phenotypic assessment of plants grown in soil, sterile seedlings were planted in Cologne soil (*Zgadzaj et al., 2016*) and the shoot biomass, together with the number of pink and total nodules were counted at 9 weeks post-planting. The soil was supplemented with 10 mM $KNO_3$ for phenotypic assessment in the presence of nitrogen.

## Acetylene reduction assays

Ethylene production was monitored in acetylene reduction assays using a SensorSense (Nijmegen, NL) ETD-300 ethylene detector as previously described (*Reid et al., 2018*).

## RT-qPCR

Root systems formed by *chit5* mutants and wild type plants were harvested 21 days after inoculation with *M. loti* R7A mock (diluted YMB medium). The plants were grown on plates supplemented with ¼ B and D medium in a 16/8 hr day/night regime of 21/16°C. The mRNA was extracted using Dynabeads (Invitrogen) and the quality of the purified mRNA was examined with a 2100 Bioanalyzer RNA pico chip (Agilent). cDNA was synthesized by RevertAid Reverse Transcriptase (Fermentas) with an OligodT primer. For quantitative real-time PCR analysis of symbiotic transcripts levels, three housekeeping genes (ATP, UBC, and PP2A encoding *L. japonicus* ATP synthase, Ubiquitin Conjugating

Enzyme and Protein Phosphatase 2A) were used as references. Specificity of the primers was ensured by melting curve analysis and sequencing of the amplification products. Quantitative real time PCR using specific primers for the reference and target genes (*Figure 1—source data 2*) was performed using the LightCycler 480 II (Roche) with the LightCycler SYBR Green I Master kit. The relative quantification software by Roche was used to determine the efficiency-corrected relative transcript levels, normalized to a calibrator sample. The geometric mean of the relative expression ratios for the three biological and three technical replicates has been calculated (*Vandesompele et al., 2002*) as well as the corresponding upper and lower 95% confidence intervals.

## Light microscopy (LM) and transmission electron microscopy (TEM) analyses including cortical infection thread (IT) counts.

For cortical ITs white nodules inoculated with ML001 DsRed were scored at four wpi. Nodules were prepared for LM and TEM investigation of infection from the indicated genotypes. Nodules that were fixed in 2.5% glutaraldehyde in 0.1 M sodium cacodylate (pH 7.0) overnight at 4°C were dehydrated in an ethanol series, and embedded in LR White acrylic resin (Agar Scientific, UK). Semithin sections (1 µm) of these were taken for light microscopy. Ultrathin (70 nm) sections were taken for TEM using a Leica UCT ultramicrotome from samples that had been post-fixed in 1% osmium tetroxide and embedded in Araldite epoxy resin. The semithin sections were collected on glass slides and stained with 0.1% toluidine blue, whereas the ultrathin sections were collected on pioloform-coated nickel grids. The number of ITs in semithin sections was counted in two representative sections of 10 different nodules for wild-type and *chit5-1* genotypes.

## Chit5 expression analyses

*Chit5* promoter region of 2072 bp upstream of start codon was PCR-amplified from *L. japonicus* Gifu using primers caccCATACTTAACCAATGTGGTACTTCAATTC (PM-9331) and GTGTATATATA TGTGAAACCTTGCATCTC (PM-9332). The amplification product was cloned into pIV10 carrying GUS or tYFP-NLS (*Reid et al., 2017*) reporters using a Gateway cloning strategy. Promoter activity was investigated in transformed roots expressing the reporter constructs at 1, 3, 7, 10, 14, 21 dpi after inoculation with *M. loti*. For analysis of *Chit5* promoter-GUS activity, roots were stained as previously described (*Kawaharada et al., 2017a*). *Chit5* promoter-tYFP-NLS activity was investigated using a Zeiss LSM780 meta confocal microscope.

## Genetic complementation

For complementation analysis, a construct consisting of *Chit5* promoter (2072 bp) followed by genomic *Chit5* sequence (3206 bp) was used. The clone was obtained by a sequential subcloning from BAC69G19 following: 1) *HindIII* digest and cloning in pGreen29, and 2) *SmaI* and *SalI* digest of pGreen29 construct, and cloning into pIV10. The construct was integrated into *Agrobacterium rhizogenes* AR1193 that was used for hairy root plant transformations. AR1193 with empty pIV10 vector was used as a control. To introduce the E166 to K mutation in CHIT5 sequence, the piV10 *Chit5* construct was used as a template for introducing the point mutation using primers GATTGGaAG TGGCCAGGAGATG (PM-11341) and CCACTtCCAATCCAAGTCAAGACC (PM-11342). The construct was then moved to *Agrobacterium rhizogenes* AR1193 for hairy root plant transformations. For *Chit5* overexpression the *Chit5* coding region from start to stop codon was amplified using pIV10-*Chit5* as template and primers caccATGATCATCAAGCTCTTGGTTGC (PM-9870) and TCAA TCATTATAAAGAGGTGAAAACAAGTG (PM-9829), followed by cloning into pIV10 vector containing the *LjUbiquitin* promoter and *Nos* terminator using Gateway cloning strategy. For *Nfr1/Nfr5* overexpression the constructs described by Radutoiu et al. (*Radutoiu et al., 2007*) have been used. All constructs have been sequence-verified.

## Root exudate hydrolysis assays

The assays were performed as described by Staehelin et al (*Staehelin et al., 1995*). with adaptation for *L. japonicus*. Surface sterilized seeds of wild-type and *chit5* mutants were germinated on upright plates with wet filter paper in a 16/8 hr day/night regime of 21/16°C for 3 days. The emerging plant roots were pre-treated with R7A Nod factor by transfer into 0.5 ml dark glass vials and grown O/N in ¼ B and D medium supplemented with 0.1 µM NodMl-V(C18:1, Me, Cb, FucAc). Pre-treated

seedlings were transferred to new 0.5 ml dark glass vials filled with 300 µl of ¼ B and D medium supplemented with 10 µM NodMI-V(C18:1, Me, Cb, FucAc) or hexa-N-acetylchitohexaose (CO- VI) for 18 hr incubation time in the dark. Seedlings were removed and the incubating solution was used for measurements of ROS induction as described by Bozsoki et al. (*Bozsoki et al., 2017*). Nod factor hydrolysis by root exudates was determined by extracting the Nod factors from the incubating solution with equal volume of distilled n-butanol for extraction. Freeze dried samples were analysed as described below.

## Analyses of Nod factor hydrolysis

The n-Butanol-extracted samples from the root exudate hydrolysis assay were dissolved in acetonitrile-water mixture (1:1 (v/v), 50 µL) and analysed on a Dionex UltiMate 3000 UHPLC$^+$ focused system. Separation of LCOs was performed using a C4 column (Phenomenex Aeris 3.6 µm widepore, 200 Å, 50 × 2.1 mm). Samples were eluted by applying a gradient of $CH_3CN$ in $H_2O$ containing 0.1% formic acid with a flow rate of 1 mL/min for 12 min. Peak integration was performed using the wavelength range 190–440 nm to maximize signal intensity. HPLC data were complemented by high-resolution mass spectrometric identification of LCO-V (NodMl-V(C18:1, Me, Cb, FucAc), and hydrolysis products LCO-III (NodMl-III(C18:1, Me, Cb)) and LCO-II (NodMl-II(C18:1, Me, Cb)) using a Bruker Impact HD UHR-QTOF mass spectrometer connected to the LC. HR-MS (ESI-TOF): calcd. for LCO-V, [M + H]$^+$: m/z 1501.7394; found 1501.7414. HR-MS (ESI-TOF): calcd. for LCO-II, [M + H]$^+$: m/z 704.4328; found 704.4346.

## Acknowledgements

We thank Clive Ronson and John Sullivan for providing the *M. loti* R7A mutants. We thank Keisuke Yokota for his contribution to fine mapping of *Chit5*. We thank Finn Pedersen for plant care. We thank Clive Ronson and Christian Staehelin for constructive discussions. We thank Jens Stougaard for critical reading of the manuscript. This work was supported by the Danish Research Foundation, DNRF 79 grant.

## Additional information

### Competing interests

Christina Kalisch: affiliated with Novozymes. The author has no other competing interests to declare. Mette U Berentsen: affiliated with Eurofins Steins Laboratorium A-S. The author has no other competing interests to declare. The other authors declare that no competing interests exist.

### Funding

| Funder | Grant reference number | Author |
| --- | --- | --- |
| Danish National Research Foundation | DNRF79 | Anna Malolepszy<br>Simon Kelly<br>Kasper Kildegaard Sørensen<br>Christina Kalisch<br>Zoltan Bozsoki<br>Michael Panting<br>Stig U Andersen<br>Dorthe Bødker Jensen<br>Maria Vinther<br>Noor de Jong<br>Lene Heegaard Madsen<br>Kira Gysel<br>Mette U Berentsen<br>Mickael Blaise<br>Knud Jørgen Jensen<br>Mikkel B Thygesen<br>Niels Sandal<br>Kasper Røjkjær Andersen<br>Simona Radutoiu |
| China Scholarship Council | 201604910506 | Ke Tao |

The funders had no role in study design, data collection and interpretation, or the decision to submit the work for publication.

## Author contributions
Anna Malolepszy, Simon Kelly, Data curation, Formal analysis, Validation, Investigation, Visualization, Methodology, Writing—original draft; Kasper Kildegaard Sørensen, Data curation, Formal analysis, Visualization; Euan Kevin James, Data curation, Formal analysis, Investigation, Visualization, Writing—original draft; Christina Kalisch, Michael Panting, Shusei Sato, Data curation, Formal analysis, Investigation; Zoltan Bozsoki, Data curation, Formal analysis, Investigation, Methodology, Writing—original draft; Stig U Andersen, Data curation, Formal analysis, Investigation, Visualization; Ke Tao, Data curation, Formal analysis, Investigation, Methodology; Dorthe Bødker Jensen, Yosuke Umehara, Mette U Berentsen, Formal analysis, Investigation; Maria Vinther, Noor de Jong, Investigation, Methodology; Lene Heegaard Madsen, Formal analysis, Investigation, Visualization, Methodology; Kira Gysel, Data curation, Investigation, Methodology; Mickael Blaise, Conceptualization, Data curation, Supervision, Methodology; Knud Jørgen Jensen, Conceptualization, Formal analysis, Supervision, Validation, Methodology; Mikkel B Thygesen, Conceptualization, Data curation, Formal analysis, Supervision, Visualization, Methodology, Writing—original draft; Niels Sandal, Data curation, Formal analysis, Supervision, Validation, Investigation; Kasper Røjkjær Andersen, Conceptualization, Data curation, Validation, Visualization, Methodology; Simona Radutoiu, Conceptualization, Resources, Supervision, Funding acquisition, Investigation, Visualization, Methodology, Writing—original draft, Project administration

## Author ORCIDs
Euan Kevin James  https://orcid.org/0000-0001-7969-6570
Zoltan Bozsoki  https://orcid.org/0000-0002-4267-9969
Stig U Andersen  http://orcid.org/0000-0002-1096-1468
Kira Gysel  http://orcid.org/0000-0003-4245-9998
Mikkel B Thygesen  https://orcid.org/0000-0002-0158-2802
Kasper Røjkjær Andersen  http://orcid.org/0000-0002-4415-8067
Simona Radutoiu  http://orcid.org/0000-0002-8841-1415

## Decision letter and Author response
Decision letter https://doi.org/10.7554/eLife.38874.030
Author response https://doi.org/10.7554/eLife.38874.031

## Additional files
### Supplementary files
• Transparent reporting form
DOI: https://doi.org/10.7554/eLife.38874.028

### Data availability
All data generated or analysed during this study are included in the manuscript and supporting files.

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
