## [Decision Letter]

Thank you for submitting your article "A plant chitinase controls cortical infection thread progression and nitrogen-fixing symbiosis" for consideration by *eLife*. Your article has been reviewed by three peer reviewers, one of whom is a member of our Board of Reviewing Editors, and the evaluation has been overseen by Christian Hardtke as the Senior Editor. The following individual involved in review of your submission has agreed to reveal his identity: J Allan Downie (Reviewer #2).

The reviewers have discussed the reviews with one another and the Reviewing Editor has drafted this decision to help you prepare a revised submission.

Summary:

The work in this manuscript demonstrates that plant chitinases in *L. japonicus* play an essential role in promoting infection of nodules by nitrogen-fixing rhizobia. It has long been thought that chitinases would be required to limit signalling induced by rhizobial Nod factors. This work shows that the *Chit5* encoded chitinases plays a specific role in promoting infection during nodule development. Additional work with bacterial mutants in conjunction with the plant chitinase mutant indicates that specific Nod-factor decorations play a special role in infection during nodule development, different from the requirements for infection thread development in root hairs.

Essential revisions:

The reviewers have agreed that the following three areas are those that are critical to address in your revision:

1) Test to see whether the *chit5* mutant plants show any growth or morphological phenotype when grown with combined nitrogen. If yes, then relate these findings to the overall conclusions of the paper.

2) Figure 1—figure supplement 2. Given that there is a trend toward higher defense gene expression in the inoculated *chit5* mutant, one cannot rule out a localized defense response as contributing to the phenotypes seen. That is, given that the authors extracted RNA from whole roots, the dilution of expression of any defense genes could obscure a defense response (which would be consistent with the trend upward). Therefore, the authors need to rule out the possibility of a localized defense response explaining the results seen. One experiment would be to conduct qRT-PCR analysis of the defense genes starting with RNA from nodules, not whole roots. However, perhaps a better experiment would be to use defense gene/promoter-GUS fusions in hairy roots to see if a localized defense response can be seen.

3) Finally, the authors interpret their analysis of the *nolL* and *nodZ* mutants as implicating a change in NF chemistry as important to the *chit5* mutant phenotypes. However, what is not addressed is that these mutations could also change the availability of the NF, either by changing concentration, hydrophobicity, or other parameter. If this were the case, then the conclusions with these mutants would be analogous to the work with the *nodD* mutants where NF concentration seemed to be most critical. The authors need to consider their data critically and modify the paper and Discussion appropriately.

---

## [Author Response]

Essential revisions:The reviewers have agreed that the following three areas are those that are critical to address in your revision:1) Test to see whether the chit mutant plants show any growth or morphological phenotype when grown with combined nitrogen. If yes, then relate these findings to the overall conclusions of the paper.

We have analysed the phenotype of WT and *chit5-3* mutant when grown in soil supplemented with 10mM KNO_3_. We found a similar overall plant phenotype, shoot and root weights were not significantly affected, indicating that *chit5* phenotypes are caused by an impairment in nitrogen fixation. This data is included in Figure 1—figure supplement 2 and in the revised text.

2) Figure 1—figure supplement 2. Given that there is a trend toward higher defense gene expression in the inoculated chit5 mutant, one cannot rule out a localized defense response as contributing to the phenotypes seen. That is, given that the authors extracted RNA from whole roots, the dilution of expression of any defense genes could obscure a defense response (which would be consistent with the trend upward). Therefore, the authors need to rule out the possibility of a localized defense response explaining the results seen. One experiment would be to conduct qRT-PCR analysis of the defense genes starting with RNA from nodules, not whole roots. However, perhaps a better experiment would be to use defense gene/promoter-GUS fusions in hairy roots to see if a localized defense response can be seen.

We thank reviewers for suggesting this possibility. We analysed the expression of these genes in the whole root because *Chit5* gene is expressed in the whole root tissue. At 21 dpi the whole root is exposed to rhizobial Nod factors and infection events take place in the root, as well as nodules. Furthermore, *chit5* mutants form 5x more nodules than wild-type that at 21 dpi these are spread throughout the whole root. In *Lotus*, the Nod factors do not elicit immune responses, however, very recent transcriptomic analyses (Kelly et al., 2018) revealed, that in wild-type plants, progression of nodule maturation is actually accompanied by a gradual downregulation of these defense markers. We took into consideration these novel results and reviewer’s suggestion and have analysed the expression of symbiotic and defense marker genes at 21 dpi specifically in nodules of wild-type and *chit5*. We did not find an activation of immune responses but did observe that similar to the early and late symbiosis genes, there is a deregulation of these defense marker genes in the *chit5* nodules compared to wild-type. Together, these results confirm that at 21 dpi, the nodules of *chit5* mutants are arrested in an earlier developmental stage when symbiosis and defense signalling are unbalanced and fail to complete their transition to functional, nitrogen-fixing organs. We have incorporated these new results in the revised manuscript (Figure 1—figure supplement 4) and amended the text accordingly.

3) Finally, the authors interpret their analysis of the nolL and nodZ mutants as implicating a change in NF chemistry as important to the chit5 mutant phenotypes. However, what is not addressed is that these mutations could also change the availability of the NF, either by changing concentration, hydrophobicity, or other parameter. If this were the case, then the conclusions with these mutants would be analogous to the work with the nodD mutants where NF concentration seemed to be most critical. The authors need to consider their data critically and modify the paper and Discussion appropriately.

In the revised text we have taken into consideration that other factors might be at play when explaining the phenotype of *chit5* inoculated with *nodZ/nolL*. The text now reads: “Compared to wild-type *M. loti*, these bacterial mutants are delayed in initiating nodule organogenesis and form fewer infected nodules (Figure 4—figure supplement 3), a phenotype similar to that induced by the *M. loti nodD1* mutant (Kelly et al., 2017). […] Alternatively, the signalling induced by

*nodZ/noL* mutants may be lower compared to wild-type bacteria and therefore may not require CHIT5-dependent modulation.”